# Covalent Functionalization of Black Phosphorus Nanosheets with Dichlorocarbenes for Enhanced Electrocatalytic Hydrogen Evolution Reaction

**DOI:** 10.3390/nano13050826

**Published:** 2023-02-23

**Authors:** Aidar M. Kuchkaev, Airat M. Kuchkaev, Aleksander V. Sukhov, Svetlana V. Saparina, Oleg I. Gnezdilov, Alexander E. Klimovitskii, Sufia A. Ziganshina, Irek R. Nizameev, Iskander R. Vakhitov, Alexey B. Dobrynin, Dmitry I. Stoikov, Gennady A. Evtugyn, Oleg G. Sinyashin, Xiongwu Kang, Dmitry G. Yakhvarov

**Affiliations:** 1Arbuzov Institute of Organic and Physical Chemistry, FRC Kazan Scientific Center of RAS, Arbuzov Street 8, 420088 Kazan, Russia; 2Alexander Butlerov Institute of Chemistry, Kazan Federal University, Kremlyovskaya Street 18, 420008 Kazan, Russia; 3Institute of Physics, Kazan Federal University, Kremlyovskaya Street 18, 420008 Kazan, Russia; 4Zavoisky Physical-Technical Institute, FRC Kazan Scientific Center of RAS, Sibirsky Tract 10/7, 420029 Kazan, Russia; 5Department of Nanotechnologies in Electronics, Kazan National Research Technical University Named after A.N. Tupolev-KAI, K. Marx Street 10, 420111 Kazan, Russia; 6New Energy Research Institute, School of Environment and Energy, South China University of Technology, Higher Education Mega Center, 382 East Waihuan Road, Guangzhou 510006, China

**Keywords:** black phosphorus, phosphorene, covalent functionalization, reactive intermediates, carbene, hydrogen evolution reaction, electrocatalysis

## Abstract

Two-dimensional black phosphorus (BP) has emerged as a perspective material for various micro- and opto-electronic, energy, catalytic, and biomedical applications. Chemical functionalization of black phosphorus nanosheets (BPNS) is an important pathway for the preparation of materials with improved ambient stability and enhanced physical properties. Currently, the covalent functionalization of BPNS with highly reactive intermediates, such as carbon-free radicals or nitrenes, has been widely implemented to modify the material’s surface. However, it should be noted that this field requires more in-depth research and new developments. Herein, we report for the first time the covalent carbene functionalization of BPNS using dichlorocarbene as a functionalizing agent. The P–C bond formation in the obtained material (BP–CCl_2_) has been confirmed by Raman, solid-state ^31^P NMR, IR, and X-ray photoelectron spectroscopy methods. The BP–CCl_2_ nanosheets exhibit an enhanced electrocatalytic hydrogen evolution reaction (HER) performance with an overpotential of 442 mV at −1 mA cm^−2^ and a Tafel slope of 120 mV dec^−1^, outperforming the pristine BPNS.

## 1. Introduction

In the past years, two-dimensional (2D) black phosphorus (BP)-based materials have received significant attention in the scientific community. Owing to their unique puckered and in-plane anisotropic structure, tunable direct bandgap, and high charge carrier mobility, black phosphorus nanosheets (BPNS) have found broad applications in various fields, such as field-effect transistors, optoelectronic and energy devices, energy conversion, biomedicine, sensors, and electro- and photo-catalysis [1,2,3,4,5,6,7,8,9]. However, the rapid degradation of BPNS in the presence of air and moisture may result in a loss of intrinsic electronic properties, which impedes the use of exfoliated BP in most practical applications [8]. In this regard, the chemical functionalization of BPNS has been proposed as an efficient method to improve the oxidation stability of the material. Moreover, besides enhanced environmental stability, chemical functionalization of BPNS has become an excellent tool for fine-tuning its intrinsic physical properties, as well as impartment of new properties of the material, and has found widespread use for the construction of new BP-based functional materials [10,11,12,13]. Currently, covalent functionalization of BPNS using diazonium or iodonium salts or organic azides as precursors for the generation of highly reactive intermediates (carbon-free radicals or nitrenes) has been extensively explored [14,15,16,17,18]. In addition, other routes for the chemical functionalization of BPNS, including reductive alkylation and functionalization with nucleophilic agents, have also been developed [19,20].

Carbenes derived from chloroform or diazirines are widely implemented for the functionalization of various carbon materials [21,22,23,24]. Chloroform, upon treatment with a strong base, forms dichlorocarbene, which is able to functionalize carbon materials by the [1 + 2] cycloaddition reaction to C=C bonds [22]. The use of diazirines as precursors for the generation of carbenes makes it possible to vary the structure of carbenes, which have been successfully used to graft the organic linkers for immobilization of metal nanoparticles and metal complexes on the surface of carbon nanotubes and graphene nanosheets [24,25]. Moreover, it was shown that the functionalization with carbene moieties might be achieved not only by the cycloaddition reactions, but also via carbene insertion reactions into C–H, N–H, or O–H groups [24,26]. The first-principal studies provided by P. Denis have shown that carbenes and nitrenes are promising reactants to functionalize phosphorene nanosheets [27]. The study revealed that the addition of dichlorocarbene and dibromocarbene to phosphorene may proceed in two ways: via a [1 + 2] cycloaddition reaction with the formation of cyclic diphosphirane fragments, and by the addition to one phosphorus atom with P=C bond formation. The latter route is more favorable as it exhibits lower reaction energy, however, in some cases, the formation of cyclic compounds upon the structure optimization has also been observed [27]. To the best of our knowledge, there are currently no experimental studies on the functionalization of BPNS with carbene intermediates. In this work, we provide a strategy for BPNS functionalization with highly reactive dichlorocarbene intermediates by treating chloroform with potassium tert-butoxide in the presence of exfoliated BP. The resulting material was used as a metal-free catalyst for the electrochemical hydrogen evolution reaction (HER).

## 2. Results and Discussion

The BPNS were obtained by the sonication-assisted liquid-phase exfoliation method in N–methylpyrrolidone (NMP). Chemical functionalization of BPNS with dichlorocarbene was performed by slow addition of a suspension of BPNS in chloroform to a suspension of potassium tert-butoxide in pentane (Figure 1). After functionalization, the obtained material (BP–CCl_2_) was washed several times with water and isopropanol, dried in vacuo, and stored under a nitrogen atmosphere. The functionalized material was then characterized by various microscopic and spectroscopic methods.

The morphological characteristics of the BPNS before and after functionalization were determined by scanning electron microscopy (SEM), transmission electron microscopy (TEM), and atomic force microscopy (AFM) methods. As shown in the AFM and TEM images (Figure 2), the exfoliated BP particles exhibit sheet-like morphology with uniform thickness and lateral size ranging from 100 to 600 nm. The thickness of BPNS estimated by the AFM method varied from 3 to 4 nm, which corresponds to 5–7 phosphorene layers. The SEM and TEM images of BP–CCl_2_ showed the retention of a sheet-like morphology and crystallinity in the functionalized material.

The Raman spectrum of bulk BP (Figure 3) shows characteristic peaks at 358.6, 432.6, and 460.2 cm^−1^, which correspond to one out-of-plane phonon mode, A^1^_g_, and two in-plane phonon modes, B^2^_g_ and A^2^_g_, of the crystalline BP. It is worth noticing the appearance of signals at 192.7 and 227.3 cm^−1^, which correspond to edge phonons B^1^_g_ and B^1^_3g_ in the crystal lattice of the BP [28]. These signals also appear in the Raman spectrum of the functionalized BP–CCl_2_ material, which indicates the retention of the overall crystallinity of the material. A slight shift of the A^1^_g_, B^2^_g_, and A^2^_g_ phonon modes to the blue region (359.9, 435.3, and 462.5 cm^−1^, respectively) indicates the ultrathin nature of the 2D material [29]. In addition, the formation of new peaks in the regions of 175, 206, 245, 273, and 298 cm^−1^, as well as the formation of signals between A^1^_g_ and B^2^_g_ phonons in the region of 385–397 cm^−1^, are also observed. The appearance of these signals, as well as the broadening of the A^1^_g_ signal, in the BP–CCl_2_ sample may indicate a distortion of the BP surface caused by the addition of a functionalizing agent. Thus, the formation of new signals in the Raman spectra in regions less than 400 cm^−1^ with especially pronounced signals in the regions of 250–300 cm^−1^ was reported for BPNS functionalized with methyl and hexyl groups [20,30]. Similar results were obtained by Sofer et al., in which the appearance of signals in the region of 230 cm^−1^ is associated with the formation of a P–C bond in BPNS functionalized with *S*-(bromomethyl) ethanethioate [19].

The magic angle spinning (MAS) ^31^P and ^31^P{^1^H} NMR spectra of BP–CCl_2_ (Figure 4) contain signals related to the BP lattice (δ = 17.81 ppm) [31] and phosphoric acids, namely, H_3_PO_3_ (δ = 6.58 ppm, ^1^J_PH_ = 688 Hz) and H_3_PO_4_ (δ = 0.36 ppm) [32]. In addition, the spectra exhibit a signal at −10.85 ppm, which may correspond to the P–CCl_2_ covalent bond of the functionalized material. The analysis of the literature data has shown that the signals in this region of the ^31^P NMR spectra refer to organic phosphines, in which the P–C bonds are realized [33,34,35], in contrast to the P=C bond, which is characterized by the signals in the regions of much lower fields [36]. In addition, similar signals in the ^31^P NMR spectrum are also observed for functionalized BP–based materials with the P–C covalent bond formation [18,37,38]. In this regard, we assume that in the BP–CCl_2_ sample, the functionalization of BPNS with dichlorocarbenes most likely proceeds via the [1 + 2] cycloaddition reaction with two phosphorus atoms of the BP lattice with the formation of a cyclic P–C–P diphosphirane derivative, in contrast to the theoretical calculations [27]. In the integration of the signals in the ^31^P{^1^H} MAS NMR spectrum, the functionalization degree of the BP–CCl_2_ sample was roughly estimated to be approximately 2.0%.

Additionally, the ^31^P{^1^H} spectrum also contains a broadened signal at −4 ppm, which we could not identify; however, the changed signal shape in the ^31^P spectrum registered without proton decoupling may be due to a signal splitting, which in turn suggests the formation of the fragments containing one or more P–H bonds. The spectra recorded in ^1^H→^31^P cross-polarization mode allow to determine the components containing hydrogen atoms at distances of one or two bonds from the corresponding phosphorus atom. Thus, the ^1^H→^31^P cross-polarization magic angle spinning (CPMAS) NMR spectrum of the BP–CCl_2_ sample exhibits signals at 6.47 and −4.92 ppm, which confirms the presence of P–H bonds in these compounds. It is worth noticing the absence of CPMAS NMR signals at −10.85 ppm, which also implies the functionalization of the material with CCl_2_ groups.

The P 2p core-level XPS spectrum of the BPNS sample (Figure 5a) exhibits characteristic lines assigned to P 2p_1/2_ and P 2p_3/2_ peaks at binding energies of 131.0 and 130.1 eV, respectively, as well as a broad peak centered at 133.9 eV, assigned to the oxidized forms of phosphorus. The low intensity of the latter peak indicates the high quality of the starting material. The high-resolution P 2p XPS spectrum of BP–CCl_2_ (Figure 5b) also exhibits the characteristic signals belonging to P 2p_1/2_ and P 2p_3/2_ peaks at binding energies of 130.7 and 129.8 eV, respectively. A comparison of P 2p_1/2_ and P 2p_3/2_ peaks of BP–CCl_2_ with the corresponding signals for pristine BPNS revealed that in the case of a functionalized material, a noticeable broadening and a shift of the doublet to the region of lower energies (by approximately 0.3 eV) are observed. These facts indicate the increased electron concentration, partial loss of long-range order, and the oxidation of P atoms on the surface of the BP–CCl_2_ sample [39,40]. A similar effect was also observed by Shao et al. for BPNS functionalized with NH_2_ groups [41]. More important is the presence of the P–C bond, which was detected at a binding energy of 133.3 eV in the P 2p spectrum of the BP–CCl_2_ sample [30]. This suggests the covalent functionalization of BPNS with the P–C bond formation.

Comparison of the high-resolution C 1s core-level XPS spectra of BPNS and BP–CCl_2_ (Figure 5c) showed a significant broadening of the signal in the case of the functionalized material, which indicates the presence of carbon-containing fragments of different compositions, apart from the substrate material. Thus, in addition to C sp^2^/sp^3^ at 284.8 eV, the C 1s spectrum of BP–CCl_2_ could be deconvoluted into components with binding energies of 284.0, 286.7, and 289.3 eV, corresponding to the C–P, C–Cl, and C=O bonds, respectively [42], which confirms the covalent functionalization of BPNS with CCl_2_ fragments.

Fourier-transform infrared spectroscopy (FT-IR) was further used to characterize the BPNS and BP–CCl_2_ samples (Figure 6). The FT-IR spectrum of BPNS does not contain clearly defined peaks, apart from the low-intensity signals related to angle and stretching vibrations of P–OH groups at 1262 and 864 cm^−1^, respectively [43], which are observed upon the partial oxidation of the material’s surface. The BP–CCl_2_ sample also exhibits peaks of P–OH groups at 1255 and 864 cm^−1^, as well as the doublet at 1020–1080 cm^−1^ assigned to stretching vibrations of P–O groups [43]. Notably, the spectrum also contains signals at 797, 701, and 662 cm^−1^ belonging to vibrations of the C–Cl bond [44] and the asymmetric and symmetric vibrations of the P–C bond, respectively [45]. These facts additionally suggest the covalent functionalization of BPNS with dichlorocarbenes with the P–C bond formation.

The electrocatalytic HER performances of BPNS and BP–CCl_2_ samples were studied using a typical three-electrode cell in a 1 M KOH electrolyte solution. A catalyst-modified glassy carbon electrode (GC) was used as a working electrode. Figure 7a shows the linear sweep voltammograms (LSV) of BPNS and BP–CCl_2_ recorded at a scan rate of 5 mV s^−1^. Note that the functionalization of the BPNS surface with dichlorocarbene leads to a significantly enhanced electrocatalytic activity in the studied process. Thus, the BP–CCl_2_ sample exhibits HER activity with an overpotential of 442 mV vs. the reversible hydrogen electrode (RHE) at a current density of −1 mA cm^–2^, which is 150 mV lower than that of the BPNS sample (592 mV). The improved catalytic activity may be due to the increased concentration of active sites for hydrogen adsorption, which are formed upon the lattice distortion of the catalytically inert basal plane of BP in the functionalized material [46]. Similar effects of increased catalytic activity in edge-rich BPNS-based materials have been observed by Wang et. al. [47] and Yu et. al. [48].

The kinetics of the electrocatalytic HER of the studied catalysts were investigated using Tafel plots (Figure 7b). The values of the Tafel slopes of BP–CCl_2_ and BPNS catalysts were 120 and 141 mV dec^−1^, respectively, which implies that for both compositions, the rate-determining step in electrochemical HER is the electrochemical hydrogen adsorption (Volmer process) [41]. A lower Tafel slope indicates faster kinetics on BP–CCl_2_ compared to pristine BPNS.

Next, using the electrochemical double-layer capacitance values (C_dl_) found by recording the CV curves at different scan rates, we estimated the electrochemical active surface area (ECSA) of the modified electrodes. Figure 7c shows the dependence of the current density: Δj = j_anode_ − j_cathode_, on the potential scan rate recorded at a potential of 50 mV (vs. RHE) for BPNS and BP–CCl_2_ catalysts. The obtained data revealed that the functionalization of the material surface significantly increases the C_dl_ values from 0.33 mF cm^−2^ for BPNS to 0.59 mF cm^−2^ for BP–CCl_2_. The corresponding ECSA is determined to be 0.69 and 0.39 cm^2^ for BP–CCl_2_ and BPNS, respectively, indicating the 1.77 times increased ECSA value in BP–CCl_2_ compared to the pristine BPNS. Thus, the BP–CCl_2_ catalyst exhibits more abundant catalytically active sites, which significantly increase its activity in HER [41].

The Nyquist diagrams obtained from the electrochemical impedance spectroscopy (EIS), as well as the corresponding equivalent circuit of BPNS, BP–CCl_2_, and GC, are shown in Figure 7d. Based on the radius of the semicircular Nyquist diagrams, the charge transfer resistances (R_ct_) of BPNS and BP–CCl_2_ catalysts were calculated to be 13.12 and 6.25 kΩ, respectively (Table 1). The lower R_ct_ for BP–CCl_2_ compared to pristine BPNS indicates a faster electron transfer rate for the BP–CCl_2_ catalyst, which further improved the catalytic activity of the material in HER.

The stability of the BP–CCl_2_ catalyst was tested by continuous CV sweeps with a scan rate of 100 mV s^−1^, from 0 to −0.6 V vs. RHE in a 1 M KOH solution. Notably, the LSV curve after 1000 cycles of CV recordings changed its morphology, showing the enhanced HER activity at a low current density (from 0 to −2.0 mA cm^−2^) and slightly decreased activity at a higher current density (ΔE = 15 mV at −5.0 mA cm^−2^). These facts suggest the changes in the material’s surface during the experiment, although they did not lead to a noticeable activity deterioration of the catalyst.

Thus, the performed electrochemical experiments indicate an improved catalytic activity of the functionalized material in the electrochemical HER. The catalytic performance of BP-CCl_2_ is comparable to other metal-free catalysts of this process (Table 2).

## 3. Materials and Methods

### 3.1. General Information

All manipulations and reactions were performed under an inert atmosphere of dry nitrogen using standard Schlenk-line or glovebox techniques. Organic solvents were distilled and stored under nitrogen before use. The commercially available reagent potassium tert-butoxide (98%, Acros Organics, Geel, Belgium) was used as purchased.

BP was obtained according to a modified procedure developed by Nilges [54]. The synthesis and characterization of BP have been published in our recent article [55]. In brief: 500 mg of red phosphorus, 364 mg of the Au/Sn alloy, and 10 mg of SnI_4_ were placed into a quartz ampoule 10 cm long and 10 mm in diameter (wall thickness 1 mm). The ampoule was sealed using a high-temperature burner in a vacuum (1 × 10^−6^ bar). Then, the ampoule was placed into a two-zone tube furnace in such a way that the temperature of the empty side of the ampoule during synthesis was maintained at approximately 50 °C lower than that of the sides with reagents. The ampoule with reagents was heated to 700 °C for 3 h. After that, the ampoule was kept at this temperature for 3 h, then slowly cooled to a temperature of 560 °C for 10 h. After cooling down, the ampoule was opened, and BP crystals formed at the empty side of the ampoule were transferred into a flask with 10 mL of toluene. The mixture was refluxed for 1 h, and after that the precipitate was decanted, dried in a vacuum (1 × 10^−6^ bar), and stored under nitrogen. The product yield was 451 mg (90%), and purity was 99.6% (according to X-ray fluorescence spectroscopy analysis).

### 3.2. Characterization Techniques

Transmission electron microscopy (TEM) images of BP nanosheets were obtained on a Hitachi HT7700 transmission electron microscope (Tokyo, Japan) at an accelerating voltage of 100 kV. Samples were deposited on 300-mesh copper grids coated with formvar/carbon support film (Agar Scientific, Essex, UK) in the case of BPNS, and 300-mesh copper grids coated with lacey carbon support film (Electron Microscopy Sciences, Hatfield, PA, USA) in the case of BP–CCl_2_.

Scanning electron microscopy (SEM) images of BP, BPNS, and BP–CCl_2_ were obtained on a Hitachi TM-1000 tabletop scanning electron microscope (Tokyo, Japan) at an accelerating voltage of 15 kV and an emission current of 35.7 mA.

The surface morphology of BP nanosheets was studied by atomic force microscopy (AFM) in ambient conditions in tapping mode on a Titanium instrument (NT-MDT, Moscow, Russia) using a standard NSG-01 silicon cantilever (NT-MDT, Moscow, Russia) with a resonant oscillation frequency of 120 kHz.

Solid-state NMR measurements were carried out on a Bruker AVANCE 400 WB NMR spectrometer (Bremen, Germany) at a temperature of T = 293 K. The measurements were carried out on the ^31^P nuclei at a frequency of 162.056 MHz. For measurements, a MAS 4 BL CP BB DVT (Bruker, Bremen, Germany) probe was used. The studied samples were packed into a 4 mm zirconium oxide rotor and were rotated at the magic angle with a rotation speed of 12 kHz. The duration of the exciting 90-degree pulses for phosphorus nuclei was 3.4 µs. The standard impulse programs zg, hpdec, and cp were used. The number of points in the time domain was the same for all experiments and was equal to 2048. The number of FID accumulations varied from 256 to 512 to obtain an acceptable signal-to-noise ratio. The time between FID accumulations in experiments on phosphorus nuclei was 20 s. The width of the spectra obtained for phosphorus nuclei was 69.5 kHz. In experiments with cross-polarization, the time varied from 1 to 5 ms to obtain the maximum signal.

X-ray photoelectron spectroscopy (XPS) measurements were performed in a UHV chamber (base pressure ∼5 × 10^−10^ mbar) equipped with an Al Kα X-ray source operated at 12.5 kV and 250 W, and a Phoibos 150 hemispherical energy analyzer (all from SPECS GmbH, Berlin, Germany). A pass energy of 100 eV (step size of 1 eV) was used for wide-range scans (survey), while 20 eV pass energies (step size of 0.1 eV, 25 scans) were used for high-resolution measurements. Peak shifts due to any apparent charging were calibrated with the carbon C 1s peak set to 284.8 eV. XPS spectra registered from experimental samples were analyzed with the CasaXPS software (Casa Software Ltd., Teignmouth, UK).

Raman spectra were collected with the Raman spectrometer SENTERRA (Bruker, Bremen, Germany). In order to register Raman spectra, linearly polarized light with a wavelength of 532 nm was used. Focusing of the laser beam on the surface of the sample was performed using a 100× objective with a numerical aperture of 0.9 (180° configuration). The laser power used in spectroscopic experiments was 2 mW. The exposure time was 60 s. Raman spectra were recorded within a spectral range of 45–3700 cm^−1^ with a resolution of 3–5 cm^−1^.

FT-IR spectra (500–4000 cm^−1^) were recorded using a Vertex 70 FT-IR spectrometer (Bruker, Bremen, Germany) equipped with an ATR accessory (MIRacle, PIKE Technologies, Madison, WI, USA). Background spectra obtained from 64 scans with a resolution of 2 cm^−1^ have been subtracted from the sample spectra.

Electrochemical studies of the HER were carried out using a BASI EC Epsilon potentiostat (West Lafayette, IN, USA) equipped with an RDE-2 cell stand in a three-electrode system, using Pt wire as an auxiliary electrode, Ag/AgCl as a reference electrode, and a catalyst-modified glassy carbon rotating disk electrode as a working electrode. To modify the electrode, a suspension consisting of 1 mg of the catalyst, 0.5 mL of isopropanol, and 10 μL of a 5% Nafion binder solution was prepared, which was dispersed by sonication for 10 min. Then, 5 µL of the resulting mixture was drop-casted to the electrode surface and dried in a vacuum. Voltammograms were recorded in 1 M KOH at a linear potential scan rate of 5 mV s^−1^ at the electrode rotation speed of 800 rpm. In all measurements, the Ag/AgCl reference electrode was calibrated with respect to the reversible hydrogen electrode (RHE) according to the formula: E_RHE_ = E_Ag/AgCl_ + 0.222 + 0.059 × pH. For the 1 M KOH medium: E_RHE_ = E_Ag/AgCl_ + 1.048 V. To estimate the electrochemically active surface area of the catalyst (ECSA), cyclic voltammograms were measured in a non-Faradaic region from 0 to 100 mV vs. RHE at scan rates of 5, 50, 100, 150, and 200 mV s^−1^. The double-layer capacitance of the system, C_dl_, was determined by plotting the dependence of the current density difference between the anode and cathode sweep: Δj = j_anode_ − j_cathode_, taken at a potential of 50 mV on the potential scan rate. C_dl_ was calculated according to the formula: C_dl_ = k/2, where k is the slope of the linear fit of this data. ECSA can be calculated as: ECSA = S (C_dl_/C_s_), where S is the electrode area (0.07 cm^2^), and C_s_ is the specific capacitance value for a flat standard with 1 cm^2^ of real surface area. The general value of C_s_ is between 20 and 60 μF cm^−2^. The C_s_ value of 60 μF cm^−2^ was used to determine the ECSA in this study.

The electrochemical impedance (EIS) spectra were recorded using CHI Electrochemical Workstation 660E (CH Instruments, Austin, TX, USA) at the equilibrium potential with the amplitude of applied sine potential of 5 mV, and the frequency varied in the range from 100 kHz to 0.1 Hz in the presence of 0.01 M K_3_[Fe(CN)_6_] and 0.01 M K_4_[Fe(CN)_6_]. The equilibrium potential was calculated as the half-sum of the cathodic and anodic peak potentials of the [Fe(CN)_6_]^3−/4−^ peak pair. The impedance parameters were calculated from the Nyquist diagrams obtained from five successive registrations of EIS spectra with the R(RC) equivalent circuit using NOVA software (version 1.11.2 Metrohm Autolab b.v., Utrecht, The Netherlands).

### 3.3. Preparation of BPNS

Here, 50 mg of BP was placed in an agate mortar and 300 μL of NMP was added. The mixture was thoroughly ground and transferred into a flask with 100 mL of NMP. The resulting suspension was poured into separate vials, which were hermetically sealed using PTFE tape. Then, the vials were sonicated (150 W, 35 kHz) for 30 h. After sonication, the suspension was centrifuged at 3000 rpm for 30 min to remove the unexfoliated BP, after which the supernatant was collected and centrifuged at 15,000 rpm. The precipitate of BPNS was washed with several portions of isopropanol by successive centrifugation–redispersion procedures (15,000 rpm). The obtained material was then transferred into a Schlenk flask, dried in a vacuum, and stored under a nitrogen atmosphere.

### 3.4. BP–CCl_2_ Synthesis

BP–CCl_2_ was obtained by the reaction of BPNS with dichlorocarbenes, which were generated from chloroform by the reaction with potassium tert-butoxide, according to the procedure proposed by Turkenburg et al. [56]. For the reaction, 10 mg of BPNS was dispersed in the mixture of 1 mL of chloroform (12.5 mmol) and 10 mL of pentane. In another flask, a suspension of 1.12 g (10 mmol) of potassium tert-butoxide in 10 mL of pentane was obtained, which was cooled to −25 °C. Then, the BPNS and chloroform suspension was slowly added in small portions to the suspension with t-BuOK. The mixture was stirred for 8 h, after which it was left overnight. The functionalized BPNS were washed with water and several portions of isopropanol by successive centrifugation–redispersion procedures (15,000 rpm), then transferred in a Shlenk flask and dried in a vacuum.

## 4. Conclusions

In conclusion, a method for BPNS functionalization with dichlorocarbene intermediates was developed. The structure and composition of the obtained BP–CCl_2_ material were elucidated and thoroughly characterized using a variety of spectroscopic methods. The obtained data suggest that the functionalization of BPNS with CCl_2_ moieties proceeds via three-membered P–C–P ring formation. This functionalization of the BPNS surface led to enhanced performance of the BP–CCl_2_ nano-catalyst in the electrochemical HER in alkaline media. It was shown that the BP–CCl_2_ material exhibited the overpotential of 442 mV at a current density of −1 mA cm^−2^ and significantly outperformed the pristine BPNS. Our study provides a new route for the preparation of BP-based functional materials using carbene precursors and promotes the development of new metal-free electrocatalysts for hydrogen evolution reactions.

## Figures and Tables

**Figure 1 nanomaterials-13-00826-f001:**
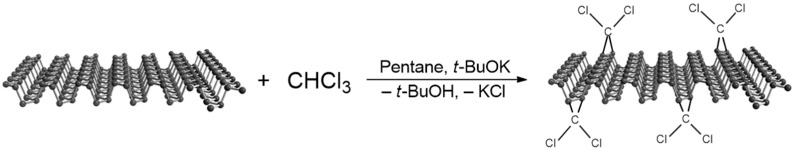
Schematic representation of BPNS functionalization with dichlorocarbene moieties.

**Figure 2 nanomaterials-13-00826-f002:**
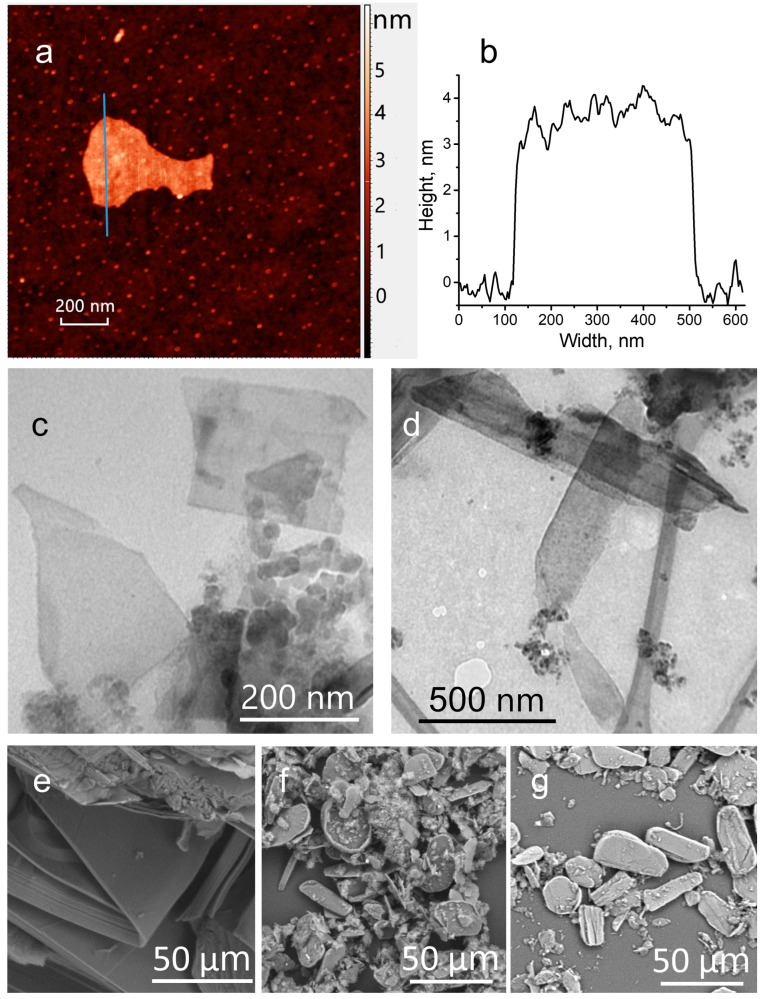
(**a**) AFM image of BPNS. (**b**) Cross-section of BPNS taken from the blue line of the AFM image. TEM images of BPNS (**c**) and BP–CCl_2_ (**d**). SEM images of BP crystal (**e**), BPNS (**f**), and BP–CCl_2_ (**g**).

**Figure 3 nanomaterials-13-00826-f003:**
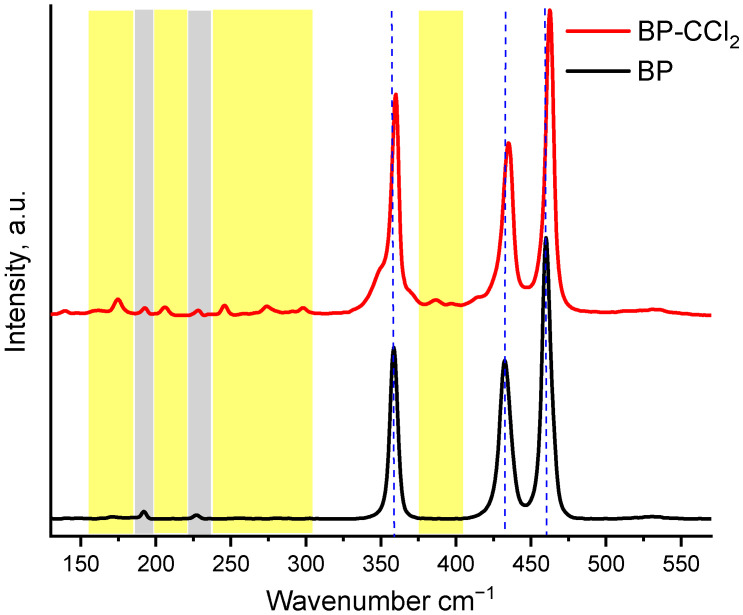
Raman spectra of BP and BP–CCl_2_. Dashed lines represent the peak positions of A^1^_g_, B^2^_g_, and A^2^_g_ phonon modes in BP, the regions of B_1g_ and B^1^_3g_ phonon modes are highlighted in gray, and the distorted vibrations of the BP lattice in BP–CCl_2_ caused by the functionalization of the material are highlighted in yellow.

**Figure 4 nanomaterials-13-00826-f004:**
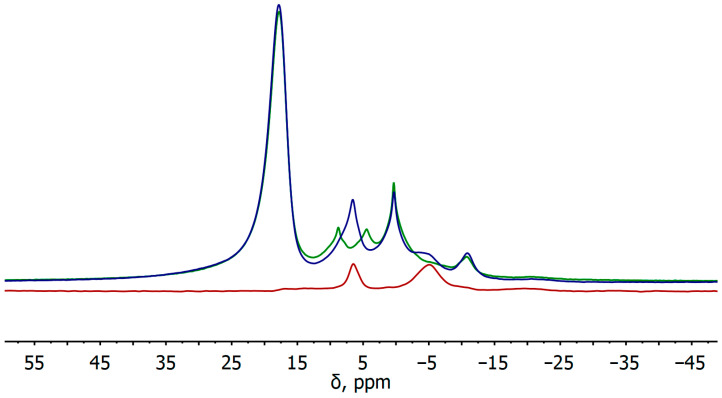
The ^31^P (green line) and ^31^P{^1^H} (blue line) MAS NMR spectra and ^1^H→^31^P CPMAS NMR spectrum (red line) of BP–CCl_2._

**Figure 5 nanomaterials-13-00826-f005:**
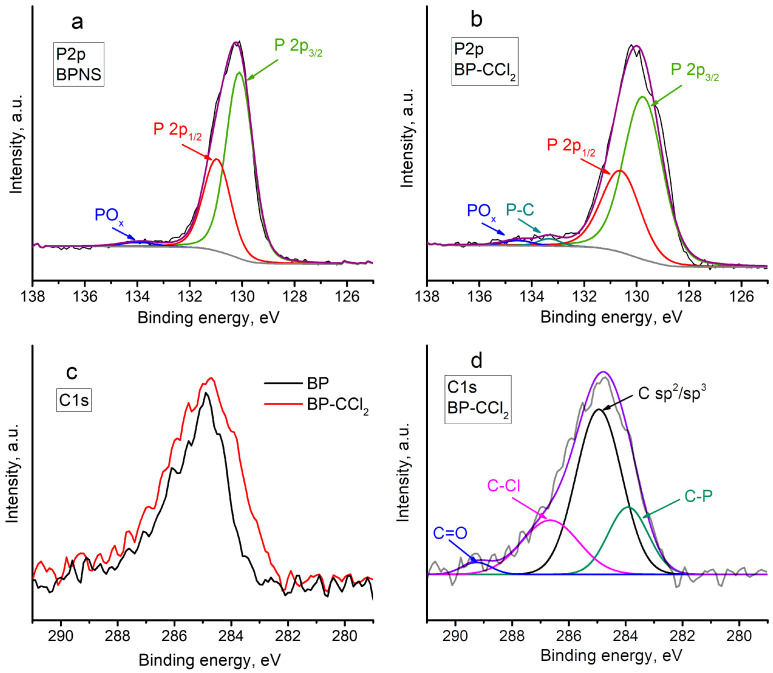
P 2p high-resolution XPS spectra of BPNS (**a**) and BP–CCl_2_ (**b**). Experimental spectra, fitting lines and background lines are shown as black, purple, and gray lines, respectively. (**c**) The comparison of the C 1s high-resolution XPS spectra of BPNS and BP–CCl_2_. (**d**) C 1s high-resolution XPS spectrum of BP–CCl_2_. Experimental spectrum and fitting line are shown as black and violet lines, respectively.

**Figure 6 nanomaterials-13-00826-f006:**
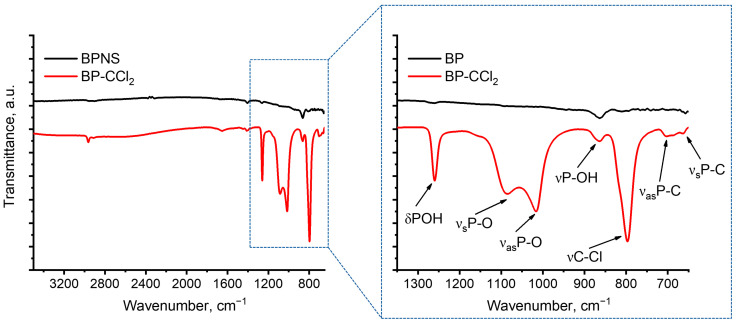
FT-IR spectra of BPNS and BP–CCl_2_. The inset contains the enlarged FT-IR spectra in the region of 650–1350 cm^−1^.

**Figure 7 nanomaterials-13-00826-f007:**
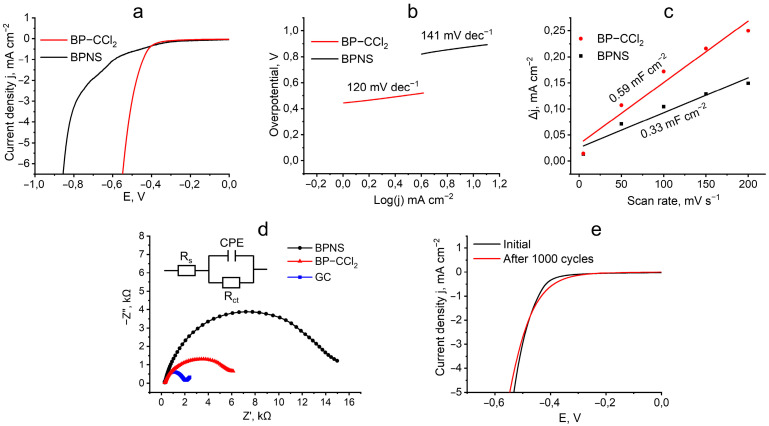
(**a**) LSV curves of BPNS and BP–CCl_2_ (vs. RHE) for the electrocatalytic HER at a scan rate of 5 mV s^−1^ in 1 M KOH. (**b**) Tafel plots for BP and BP–CCl_2_. (**c**) The fitting curve of BPNS and BP–CCl_2_ between scan rate and current density variation (Δj = j_anode_–j_cathode_) at 50 mV (vs. RHE). (**d**) Nyquist diagrams of BPNS, BP–CCl_2_, and glassy carbon electrode. The inset shows the equivalent circuit consisting of solution resistance, R_s_, charge-transfer resistance, R_ct_, and constant phase element (CPE). (**e**) LSV curves of BP–CCl_2_ initially and after 1000 cycles of CV recordings.

**Table 1 nanomaterials-13-00826-t001:** The EIS parameters (R_s_, R_ct_, CPE, and the value of the exponent of the CPE (n)) obtained by fitting Nyquist diagrams using the equivalent circuit.

Catalyst	R_s_, Ω	R_ct_, kΩ	CPE, μF	n
BPNS	188.4 ± 1.7	13.12 ± 0.29	4.02 ± 0.23	0.994 ± 0.002
BP–CCl_2_	177.8 ± 2.5	6.25 ± 0.21	3.48 ± 0.18	0.992 ± 0.001
GC	190.6 ± 1.3	1.84 ± 0.11	0.89 ± 0.06	0.997 ± 0.001

**Table 2 nanomaterials-13-00826-t002:** Electrocatalytic HER performance of different metal-free catalysts.

Catalyst	Overpotential (vs. RHE), mV	Tafel Slope, mV dec^−1^	Electrolyte	Ref.
BP	1100 at 10 mA cm^−2^	212	0.5 M H_2_SO_4_	[49]
BPNS	740 at 5 mA cm^−2^	182	0.5 M H_2_SO_4_	[50]
Large BP sheets	615 at 20 mA cm^−2^	-	0.5 M H_2_SO_4_	[47]
Small BP sheets	511 at 20 mA cm^−2^	-		
NH_2_-BP	290 at 10 mA cm^−2^	63	1 M KOH	[41]
N,S-graphitic sheets	310 at 10 mA cm^−2^	112	0.1 M KOH	[51]
N-doped graphene	337 at 10 mA cm^−2^	157	0.1 M KOH	[52]
Few-layer BP (FL-BP)	618 at 10 mA cm^−2^	249	0.5 M H_2_SO_4_	[53]
Reduced graphene oxide (rGO)	623 at 10 mA cm^−2^	257		
B-doped graphene (BG)	523 at 10 mA cm^−2^	318		
FL-BP@rGO	518 at 10 mA cm^−2^	210		
FLBP@BG	386 at 10 mA cm^−2^	110		
BP–CCl_2_	442 at 1 mA cm^−2^	120	1 M KOH	This work

## Data Availability

The data presented in this study are contained within the article or are available upon request from the corresponding author, Dmitry G. Yakhvarov.

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
