# Peer review of "Covalent Functionalization of Black Phosphorus Nanosheets with Dichlorocarbenes for Enhanced Electrocatalytic Hydrogen Evolution Reaction"

_nanomaterials, 2023, doi:10.3390/nano13050826_

Round 1

Reviewer 1 Report

The authors reported the covalent carbene functionalization of BPNS using dichlorocarbene as a functionalizing agent. The P–C bond formation in the obtained material (BP–CCl2) has been confirmed by Raman, solid-state 31P NMR, XPS, and IR spectroscopy methods. The BP–CCl2 nanosheets exhibit enhanced electrocatalytic hydrogen evolution reaction (HER) performance with an overpotential of 442 mV at –1 mA cm–2 and Tafel slope of 120 mV dec–1, outperforming the pristine BPNS. Although being interesting, I find that there are some major issues with the paper that require addressing prior to this being considered for publication in this journal. I have identified the main points for consideration below:

1.     This manuscript has some spelling typos, style errors and grammatical errors, which severely affect its readability. So, I suggest the authors carefully check the whole manuscript and correct them.

2.     The ambient stability of dichlorocarbene functionalized-BPNS should be investigated. In addition, I suggest that the authors make a comparison on the ambient stability between the dichlorocarbene functionalized-BPNS and other functionalized ones.

3.     I suggest the authors make an object comparison on the HER performance between the dichlorocarbene functionalized-BPNS and other 2D nanomaterials.

4.     I suggest the authors investigated the EIS plot of the functionalized and pristine BPNS.

5.     Why did the dichlorocarbene functionalized-BPNS improve the HER performance? I suggest the author make a detail explanation about it.

6.     In the introduction section, recent studies related to BPNS functionalization should be cited, for example, Journal of Hazardous Materials 436 (2022) 129107; Materials Today Chemistry 26 (2022) 101043.

Author Response

Reviewer #1

The authors reported the covalent carbene functionalization of BPNS using dichlorocarbene as a functionalizing agent. The P–C bond formation in the obtained material (BP–CCl2) has been confirmed by Raman, solid-state 31P NMR, XPS, and IR spectroscopy methods. The BP–CCl2 nanosheets exhibit enhanced electrocatalytic hydrogen evolution reaction (HER) performance with an overpotential of 442 mV at –1 mA cm–2 and Tafel slope of 120 mV dec–1, outperforming the pristine BPNS. Although being interesting, I find that there are some major issues with the paper that require addressing prior to this being considered for publication in this journal. I have identified the main points for consideration below:

  1. This manuscript has some spelling typos, style errors and grammatical errors, which severely affect its readability. So, I suggest the authors carefully check the whole manuscript and correct them.

Response: The manuscript was carefully checked and the errors have been corrected.

  1. The ambient stability of dichlorocarbene functionalized-BPNS should be investigated. In addition, I suggest that the authors make a comparison on the ambient stability between the dichlorocarbene functionalized-BPNS and other functionalized ones.

Response: The main goal of this study was to explore the reactivity of BP nanosheets towards carbene intermediates, rather than the preparation of new BP-based materials with enhanced ambient stability. This is why the environmental stability of the material was not specifically investigated. Nevertheless, we have registered the Raman spectrum of one of our samples after two weeks of air exposure. The preservation of the characteristic signals of black phosphorus in the spectrum indicated the maintained crystallinity of the material. However, this experiment was only qualitative, not quantitative, so it was difficult to estimate both the degradation degree of the material and its overall oxidation stability. Unfortunately, we cannot perform additional stability tests due to a short period of time allowed for the revisions.

  1. I suggest the authors make an object comparison on the HER performance between the dichlorocarbene functionalized-BPNS and other 2D nanomaterials.

Response: A table with the comparison of HER performance of BP-CCl2 and other metal-free catalysts has been added to the main body text of the manuscript.

  1. I suggest the authors investigated the EIS plot of the functionalized and pristine BPNS.

Response: The data of EIS experiments have been added to the paper.

  1. Why did the dichlorocarbene functionalized-BPNS improve the HER performance? I suggest the author make a detail explanation about it.

Response: The explanation regarding improved catalytic activity of the functionalized material has been added to the main body text of the manuscript. In the HER, the catalytic activity of BP is provided by the edges and defects of the material, while the basal plane of BP is practically not involved in the catalytic process. We assume that the functionalization of the BP surface leads to the lattice distortion of the catalytically inert basal plane of BP in the functionalized material, which in turn leads to the formation of new catalytically active centers.

  1. In the introduction section, recent studies related to BPNS functionalization should be cited, for example, Journal of Hazardous Materials 436 (2022) 129107; Materials Today Chemistry 26 (2022) 101043.

Response: We thank the reviewer for this suggestion. The mentioned articles have been cited in the introduction part.

Reviewer 2 Report

The author describes the “Covalent functionalization of black phosphorus nanosheets with dichlorocarbene for enhanced electrocatalytic hydrogen evolution reaction”. This paper is quite interesting from a technological point of view. The author should revise their manuscript based on the comments and suggestions.  I recommended a Major revision of the manuscript.

The Major suggestion below:

  1. Why did the author use dichlorocarbene as a functionalizing agent, is there any particular reason?
  2. The figure quality is too poor and the author should improve the quality of the images. As well as all figure formats are different, the author should check clearly.
  3. What is the main advantage of functionalized BPNS using HER?
  4. The author should provide the prepared catalysts' electrochemically active surface area (ECSA).
  5. The author should provide the IR-corrected LSV curves and compare them with the original LSV.
  6. The author should compare your result with the previous result. Please provide the comparison table.
  7. The author should provide the SEM and TEM images of BPNS and BP–CCl2.

Author Response

Reviewer #2

The author describes the “Covalent functionalization of black phosphorus nanosheets with dichlorocarbene for enhanced electrocatalytic hydrogen evolution reaction”. This paper is quite interesting from a technological point of view. The author should revise their manuscript based on the comments and suggestions.  I recommended a Major revision of the manuscript.

The Major suggestion below:

  1. Why did the author use dichlorocarbene as a functionalizing agent, is there any particular reason?

Response: To date, most of the methods of BP functionalization with organic substrates are based on the use of precursors of high reactive intermediates as functionalizing agents. Namely, diazonium and iodonium salts are used to functionalize BP with carbon free radicals, and organic azides are used for BP functionalization with nitrene moieties. The development of new methods for the functionalization of black phosphorus using other high reactive intermediates is of great interest. To the best of our knowledge, there are currently no experimental studies on the functionalization of BP nanosheets with carbene intermediates. However, the theoretical calculations revealed the high activity of carbene intermediates towards phosphorene sheets. This is why we decided to develop a method of BP functionalization with carbenes. Dichlorocarbene was chosen as the functionalizing agent because it is the easiest to generate, as well as it has been successfully used to functionalize various carbon materials, including graphene sheets.

  1. The figure quality is too poor and the author should improve the quality of the images. As well as all figure formats are different, the author should check clearly.

Response: We changed some of the figures to improve the quality of the images. Additionally, we ask the editors to check if the image quality is sufficient and let us know if any changes are required. We could also send the original images in case of needs.

  1. What is the main advantage of functionalized BPNS using HER?

Response: The main advantages of our BPNS-based material, as well as other metal-free hydrogen evolution catalysts is their earth-abundance and low cost compared to noble-metal-based catalysts, as well as ease of processing and utilization.

  1. The author should provide the prepared catalysts' electrochemically active surface area (ECSA).

Response: The estimated values of ECSA have been added to the article.

  1. The author should provide the IR-corrected LSV curves and compare them with the original LSV.

Response: Due to high conductivity of working solution the applied IR correction didn't cause any significant changes in the electrochemical characteristics of the studied materials.

  1. The author should compare your result with the previous result. Please provide the comparison table.

Response: A table with the comparison of HER performance of BP-CCl2 and other metal-free catalysts has been added to the paper.

  1. The author should provide the SEM and TEM images of BPNS and BP–CCl

Response: SEM images of BP, BPNS and BP-CCl2, as well as TEM image of BP-CCl2 particles have been added to the main text of the manuscript (see Figure 2).

Reviewer 3 Report

This manuscript presents a Covalent functionalization of black phosphorus nanosheets with dichlorocarbenes for enhanced electrocatalytic hydrogen evolution reaction. The manuscript may be published after the authors address the following points properly.

 Comments:

1.     The author needs to use a form of the word before abbreviating (Example= Line no: 26–27; NMR, XPS, and IR)

2.     In Figure 7, all the figure x and y axis legends should be in dot type (0.1, 0.2…), not comma type.

3.     Authors need to change "sweep rate" to "scan rate" (Line no: 220)

4.     Line no: 220 should say "50 mV s-1" rather than "50 mV ".

5.     The authors must provide an EIS study for the proposed material.

6.     The authors need to provide the proposed electrode material's limitations.

7. Some grammatical, syntax, word usage, or formatting errors still exist in the manuscript.

Author Response

Reviewer #3

This manuscript presents a Covalent functionalization of black phosphorus nanosheets with dichlorocarbenes for enhanced electrocatalytic hydrogen evolution reaction. The manuscript may be published after the authors address the following points properly.

Comments:

  1. The author needs to use a form of the word before abbreviating (Example= Line no: 26–27; NMR, XPS, and IR)

Response: NMR and IR are commonly used abbreviations and usually don’t need to be defined. However, we defined XPS, AFM, TEM and other abbreviations in the text.

  1. In Figure 7, all the figure x and y axis legends should be in dot type (0.1, 0.2…), not comma type.

Response: The figures have been corrected.

  1. Authors need to change "sweep rate" to "scan rate" (Line no: 220)

Response: The phrase has been corrected.

  1. Line no: 220 should say "50 mV s-1" rather than "50 mV ".

Response: 50 mV is correct, as the measurement was taken at that potential. The double-layer capacitance of the system Cdl was determined by plotting the dependence of the current density difference between the anode and cathode sweep Δj=ja–jc taken at a potential of 50 mV on the potential scan rate. Nevertheless, we have rephrased the figure caption to make it clearer.

  1. The authors must provide an EIS study for the proposed material.

Response: The data of EIS experiments have been added to the paper.

  1. The authors need to provide the proposed electrode material's limitations.

Response: The main limitation of the material is its moderate catalytic activity. Even though BP-CCl2 exhibited activity comparable to other metal-free catalysts, for practical applications more active catalysts are required.

  1. Some grammatical, syntax, word usage, or formatting errors still exist in the manuscript.

Response: The manuscript was carefully checked and the errors have been corrected.

Round 2

Reviewer 1 Report

Accept in present form.

Reviewer 2 Report

I recommended acceptance for the publication.

Reviewer 3 Report

Accept in the current form.